# Impact of regenerative procedure on the healing process following surgical root canal treatment: A systematic review and meta-analysis

**Nader Muthanna**[1,2‡], **Xiaoyue Guan**[1,2‡], **Fouad Alzahrani**[3], **Badr Sultan Saif**[4], **Abdelrahman Seyam**[1,2], **Ahmed Alsalman**[1,2], **Ahmed Es Alajami**[5], **Ang Li**[1,6]*

1 Key Laboratory of Shaanxi Province for Craniofacial Precision Medicine Research, College of Stomatology, Xi'an Jiaotong University, Xi'an, Chin, 2 Department of Endodontic, College of Stomatology Xi'an Jiaotong University, Xi'an, China, 3 Pulp Biology and Endodontic Department, Al-Baha Dental Center, Al-Baha, Saudi Arabia, 4 Department of Orthodontics, College of Stomatology, First Affiliated Hospital of Xi'an Jiaotong University, Xi'an, China, 5 Department of Oral Preventive, College of Stomatology Xi'an Jiaotong University, Xi'an, China, 6 Department of Periodontology, College of Stomatology, Xi'an Jiaotong University, Xi'an, China

‡ NM and XG contributed equally to this work and are considered joint first authors.
* drliang@mail.xjtu.edu.cn

**Data Availability Statement:** All relevant data are within the manuscript and its Supporting Information files.

## Abstract

### Introduction

Different Guided Tissue Regeneration (GTR) procedures, such as membranes, bone substitute materials, and Autologous Platelet Concentrates (APCs), have been applied after surgical root canal treatment (SRCT), which produce different outcomes. This study aimed to evaluate the impact of regenerative procedures on the healing process following SRCT.

### Methods

A comprehensive search of PubMed, Embase, Scopus, Cochrane, and the Web of Science found Randomized Controlled Trials (RCTs) published until February 25, 2024. Manual searches were also conducted. Our main outcome was SRCT success or failure after GTR procedures. The Risk Ratio (RR) and failure rate meta-analysis used a fixed effects model with a 95% confidence interval (CI). Subgroup analyses were conducted based on the use of different GTR procedures for varying lesion types in SRCT.

### Results

Out of 1,605 records, 16 studies with 690 lesions were included. Overall, GTR procedures significantly improved healing after SRCT in both 2D (RR: 0.50; 95% CI, 0.34–0.73; P < 0.001) and 3D evaluation methods (RR: 0.36; 95% CI, 0.15–0.90; P < 0.001) with no significant difference between the two methods.

**Funding:** The author(s) received no specific funding for this work.

**Competing interests:** The authors have declared that no competing interests exist.

## Conclusion

GTR significantly improved SRCT healing regardless of the evaluation method used. Combining collagen membranes with bovine bone-derived hydroxyapatite significantly enhanced the healing process. Additionally, GTR procedures significantly improve healing in through-and-through lesions.

## Introduction

Bacterial infection can result in pulpal inflammation, ultimately leading to pulp necrosis and periapical lesions [1]. While conventional Root Canal Treatment (RCT) is the first-line treatment, Surgical Root Canal Treatment (SRCT) is recommended in cases where non-surgical RCT is unsuccessful [2, 3]. The main objective of SRCT is to create an optimal environment for periapical tissue repair. This is typically achieved by eliminating infections and inaccessible areas within the root canal system and preventing future infections [4].

Success in both RCT and SRCT relies on the absence of signs of infection and inflammation, along with radiography showing reduced periapical lesion size and normal growth of the periodontal ligament gap [5]. The evaluation of healing after SRCT is commonly conducted using the criteria established by Rud et al.[6] and Molven et al. on 2D imaging [7]. On the other hand, the Modified PENN 3D criteria have been used to evaluate healing on 3D imaging[8].

Wound healing after SRCT can result in repair or regeneration. Repair involves the development of new tissue that differs from the original cells, while regeneration involves wound healing using cells from a similar tissue [9]. However, the potential for connective tissue to invade the bony defect can interfere with the healing process [10]. Guided Tissue Regeneration (GTR) procedures have shown promise in periodontology and dental implants and have gained interest as a supplementary approach for SRCT to improve healing and prevent soft tissue collapse within the bony defect [11, 12].

Different materials for GTR can be used in SRCT, including barrier membranes, bone grafts, and Autologous Platelet Concentrates (APCs), either alone or in combination [4]. The first commercially produced material was expanded polytetrafluoroethylene (e-PTFE), but its complete removal requires additional surgery [13]. Resorbable membranes, like collagen membranes, were developed in the 1990s to avoid the need for surgical removal [14].

Different types of bone grafts offer unique advantages, disadvantages, and success rates. Familiar sources include autogenous, xenograft, allograft, and alloplast [12]. The gold standard for bone grafting is autogenous grafts for their ability to promote osteogenesis, osteoinduction, and osteoconduction characteristics [15]. However, they have disadvantages such as longer surgical time, morbidity, and limited bone supply [16]. Xenograft bone, taken from animals like bovines, is becoming more popular for its osteoconductive properties [17, 18]. Allograft bone, donated between genetically dissimilar individuals, can be osteoconductive or osteoinductive without additional surgery. Alloplasts are synthetic materials considered exclusively osteoconductive [19].

GTR using APCs showed high amounts of cytokines and growth factors, making it a promising option for tissue regeneration [20]. These cells are obtained from the patient's peripheral blood, making the procedure safer, well-tolerated, and cheaper [20]. Meanwhile, platelet-rich plasma (PRP), the first generation of APCs, is difficult to prepare and requires anticoagulants [21–23]. The second generation of platelet-rich fibrin (PRF) can be obtained through a single

centrifugation method [24]. A third-generation injectable PRF (i-PRF) was developed in 2014 using a different centrifugation force and plastic tubes, reducing clotting time [25].

Many studies have evaluated the efficacy of GTR procedures[26–28]. However, there is still debate about their impact on improving success rates after SRCT. Therefore, a systematic review and meta-analysis of these studies are necessary to aid clinicians in making informed decisions for successful SRCT. This study aimed to evaluate the impact of GTR procedures on the healing process following SRCT.

## Material and methods

The systematic review and meta-analysis followed PRISMA 2020 [29] and the Cochrane Handbook for Systematic Reviews of Interventions [30]. The systematic review process was registered in PROSPERO (CRD 42023477089). The PICOST strategy was utilized to formulate the clinically relevant question: Among individuals having Surgical Root Canal Treatment (P), will the application of a GTR procedure (I) versus not using a GTR procedure (C) have an impact on the healing process (O) in randomized clinical trials (S) after one year follow up (T)?

### Search strategy

The online search was carried out independently by three researchers (N.M., X.G., and A.S.) to locate relevant studies. The electronic databases PubMed, Embase, Scopus, Cochrane, and Web of Science were searched until February 25, 2024. Grey literature was found using Google Scholar and OpenGrey. No restrictions were placed on the publication date or language. Specific keywords were merged using Boolean operators, and the MeSH terms were incorporated into the electronic search strategy (S1 Table). In addition, manual searches were also conducted by checking the reference lists of relevant articles.

### Inclusion and exclusion criteria

This systematic review compared randomized clinical trials that used GTR procedures in the intervention group to a control group that did not. The trials had to have a minimum follow-up of 12 months and focus on periapical lesions caused by endodontic problems in human patients. Clinical assessment was based on signs and symptoms, while radiographic assessment used criteria established by Molven et al. [7] or Rud et al. [6] for 2D imaging, and modified PENN 3D criteria by Schloss et al. [8] for 3D imaging. Eligible patients had to be classified as American Society of Anesthesiologists (ASA) I or II. Studies were excluded if they did not have sufficient data, included patients with root fractures, resorption, or perforation, or included children under 12 or with sample sizes less than 10.

### Study selection

The records were imported into EndNote X21 (Clarivate Analytics, Philadelphia, PA). After the removal of duplicates, the titles and abstracts of the remaining records were screened independently for eligibility by 3 reviewers (N.M., X.G., and B.S.). The same three authors read the full texts of articles independently to determine if they met the inclusion criteria. References of all relevant research were also checked. A fourth reviewer (A.L.) was consulted to facilitate compromise in any disagreement.

### Data extraction

Three independent reviewers (N.M., X.G., and F.A.) Obtained data from every study using a standardized Excel spreadsheet. The form included the following information for each study:

the first author's name, the year of publication, the age, the size or type of lesion, the regenerative techniques and materials used, the sample size, and the outcomes observed at the 12-month follow-up. Studies with missing outcome data were excluded.

The clinical outcomes were evaluated by the presence or absence of signs of infection and inflammation. Radiographically, the healing assessment was determined by using the criteria established by Rud et al. [6] or Molven et al. [7] (complete, incomplete, uncertain, or unsatisfactory healing) for 2D imaging evaluation, whereas the modified PENN 3D criteria established by Schloss et al. [8] (complete, limited, uncertain, or unsatisfactory healing) were used for 3D imaging evaluation.

The assessment of success and failure was determined based on a comprehensive evaluation of both clinical and radiological outcomes. For statistical purposes, the outcomes were also dichotomized into success and failure. The success was referred to the loss of clinical symptoms and the signs of (complete or incomplete healing) for 2D imaging and (Complete or Limited healing) for 3D imaging. Failure was referred to the presence of clinical symptoms and/or the occurrence of (uncertain or unsatisfactory healing) for 2D and 3D imaging.

## Quality assessment

N.M., B.S., and F.A. evaluated the risk of bias in each study using the Cochrane Collaboration tool for randomized trials (RoB 2). They assessed trials using RoB 2 questions and determined each study's risk of bias using the algorithm in the RoB 2 guidance. Each study's overall risk of bias was assessed by considering each domain's risk. If all domains of the study were low risk, the study was evaluated as having a low risk of bias overall. If the study had some concerns in at least one domain but no high risk, the study was evaluated as having some concerns of bias overall. If the study had a high risk in one or more domains, it was considered to have a high risk of bias overall. Disagreements amongst reviewers were resolved through discussion. Otherwise, a fourth reviewer, A.L., was consulted until agreement was reached.

## Statistical analysis

The Cochrane Collaboration System's Review Manager 5.4 was utilized to calculate the risk ratio (RR) to compare SRCT failures with and without GTR treatments. The chi-squared test (X2) assessed the study's heterogeneity. Since the heterogeneity was small, the fixed-effects model was used (p > 0.1 or I2 ≤ 50%). Funnel plots were used to evaluate publication bias.

## Certainty of evidence assessment

GRADEpro Guideline Development Tool software (Evidence Prime, Inc, Seattle, WA) was used to create a summary of the findings table to assess the strength of the evidence. Individual GRADE criteria were considered, and evidence certainty was calculated. The GRADE system evaluates evidence certainty as high, moderate, low, or very low [31].

# Results

## Study selection

EndNote X21 received 1605 records from different electronic databases. After removing the duplicate publications, a total of 1273 were excluded based on their title and abstract, resulting in 30 papers being considered for full-text review. All studies identified after excluding duplications are shown in S2 Table. In parallel, 4 references were identified through manual searches. Next, 34 papers were full-text reviewed for eligibility, 18 studies [32–49] were excluded for different reasons (S3 Table). Finally, 16 studies [50–65] met the inclusion criteria

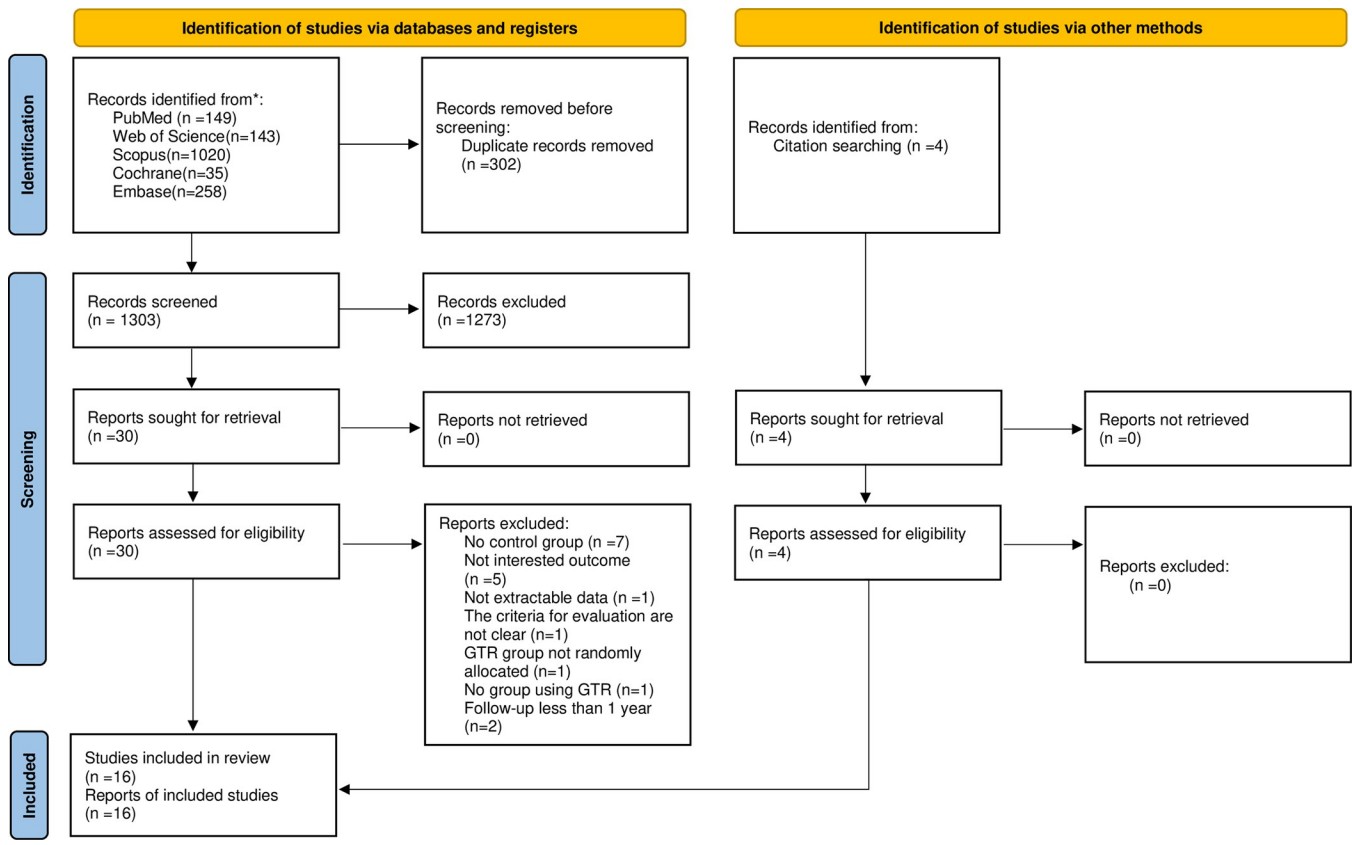

**Fig 1. PRISMA statement 2020 flow diagram.**

and were included in the meta-analysis (Fig 1). Three investigators, N.M., X.G., and B.S., conducted all the search processes and examined all the studies separately. In cases of disagreement, a fourth reviewer, A.L., was consulted in order to reach an agreement.

## Study characteristics

All data extracted in primary studies is shown in S4 Table. Table 1 summarizes the main characteristics of each of the included studies. All papers included in the analysis were RCTs published after 2000, except for Pecora et al. [59]. The intervention groups showed 381 lesions, whereas the control groups had 309 lesions. All studies included a control group that did not use GTR procedures and an experimental group that utilized GTR. All studies analyzed 2D radiographic healing based on the criteria defined by Molven et al. or Rud et al. [6, 7], and four studies [50, 51, 53, 57] investigated 3D radiographic healing using the modified PENN 3D criteria.

## Quality assessment

The risk of bias in all trials is presented in (Fig 2). However, the randomization process is not well explained in seven studies [52, 55, 56, 58, 59, 64, 65]. Deviations from intended interventions and missing outcome data were assessed as low risk for all studies. Outcome measurement was considered to have some concerns in six studies [52, 55, 56, 59, 64, 65]. Bias due to selective reporting was considered some concern in twelve studies [52, 54–56, 58–65]. Overall, four studies [50, 51, 53, 57] show a low risk of bias, and the remaining twelve studies [52, 54–

**Table 1. Characteristics of included studies.**

| Author/year | Lesion type | GTR | | Intervention group (Ex. G) | | | Control group (Cont. G) | | |
|---|---|---|---|---|---|---|---|---|---|
| | | Technique | Materials | Cases | success | Failure | Cases | Success | Failure |
| Albagle et al. 2023 [50] | Confined | BG | Collagen-based BG | 32 | 29 | 3 | 26 | 24 | 2 |
| Arpitha M et al. 2023 [51] | Through and through | BG + APCs | Collagen-based BG mixed with i-PRF | 18 | 18 | 0 | 16 | 14 | 2 |
| Chen &Shen 2016 [52] | Confined | BG + Mb | Bovine bone Hap + Collagen Mb. | 17 | 17 | 0 | 19 | 18 | 1 |
| | Through and through | | | 25 | 23 | 2 | 19 | 13 | 6 |
| Dhamija et al. 2020 [53] | Through and through | APCs | PRP | 16 | 15 | 1 | 16 | 15 | 1 |
| Dhiman et al. 2015 [54] | Apico-marginal confined to the buccal aspect of the root | APCs | PRF as Mb | 15 | 13 | 2 | 15 | 12 | 3 |
| Dominiak et al. 2009 [55] | Mean = width 8.38mm, height 9mm, depth 8.40 mm3(Confined) | Mb | Collagen Mb. | 26 | 21 | 5 | 25 | 16 | 9 |
| | | BG | Bovine-derived Hap | 30 | 25 | 5 | | | |
| | | BG + APCs as Mb | Bovine-derived Hap + PRP | 25 | 23 | 2 | | | |
| Pan et al. 2011 [56] | EX. G = (14.77 +- 4.62 mm), Cont. G = (13.31+-5.51 mm) | BG + Mb | Bovine bone Hap + Collagen Mb. | 22 | 21 | 1 | 21 | 19 | 2 |
| Parmar et al. 2019 [57] | D≥10mm &through and through | Mb | Collagen Mb. | 15 | 15 | 0 | 15 | 14 | 1 |
| Pecora et al. 1995 [59] | D≥10mm &through and through | Non-resorbable Mb | e-PTFE Mb | 10 | 10 | 0 | 10 | 10 | 0 |
| Pecora et al. 2001 [58] | D≥10mm &through and through | BG | Calcium sulfate | 10 | 9 | 1 | 10 | 8 | 2 |
| Rohilla et al. 2022 [60] | Apico-marginal confined to the buccal aspect of the root | Mb | Collagen Mb. | 12 | 10 | 2 | 8 | 8 | 0 |
| Taschieri et al. 2007 [61] | 4 wall Defect(confined) | BG + Mb | Bovine bone Hap + Collagen Mb. | 16 | 14 | 2 | 22 | 18 | 4 |
| | D≥10mm &through and through | | | 8 | 6 | 2 | 13 | 8 | 5 |
| Taschieri et al. 2008a [62] | D≥10mm &through and through | BG + Mb | Bovine bone Hap + Collagen Mb. | 17 | 15 | 2 | 14 | 8 | 6 |
| Taschieri et al. 2008b [63] | 4 wall Defect (confined) | BG + Mb | Bovine bone Hap + Collagen Mb. | 16 | 14 | 2 | 22 | 18 | 4 |
| | D≥10mm &through and through | | | 17 | 15 | 2 | 14 | 9 | 5 |
| Tobon et al. 2002 [64] | 0.04–506 mm2 (confined) | Non-resorbable Mb | e-PTFE Mb | 9 | 7 | 2 | 9 | 8 | 1 |
| | | BG + Mb | Synthetic bioactive graft Hap + e-PTFE | 8 | 8 | 0 | | | |
| Wang et al. 2017 [65] | 25–640 mm2 | BG + Mb | Bovine bone Hap + Collagen Mb. | 17 | 15 | 2 | 15 | 11 | 4 |

GTR, Guided tissue regeneration; Ex. G, Experimental group; Cont. G, Control group; BG: Bone Graft; Mb, Membrane; APCs, Autologous Platelet Concentrates; Hap, Hydroxyapatite; e-PTFE, expanded polytetrafluoroethylene; i-PRF, injectable platelet-rich fibrin; PRF, platelet-rich fibrin; PRP, platelet-rich plasma

56, 58–65] exhibit some concerns regarding risk of bias. Completed risk of bias assessments are shown in S5 Table.

## Meta-analysis

Due to the lack of noticeable heterogeneity within the studies included, the meta-analysis used a fixed-effects model. The meta-analysis for all included studies reported the failure rate according to 2D evaluation. The results showed that using the GTR following SRCT significantly improved the healing process compared to the conventional SRCT (RR: 0.50; 95% CI,

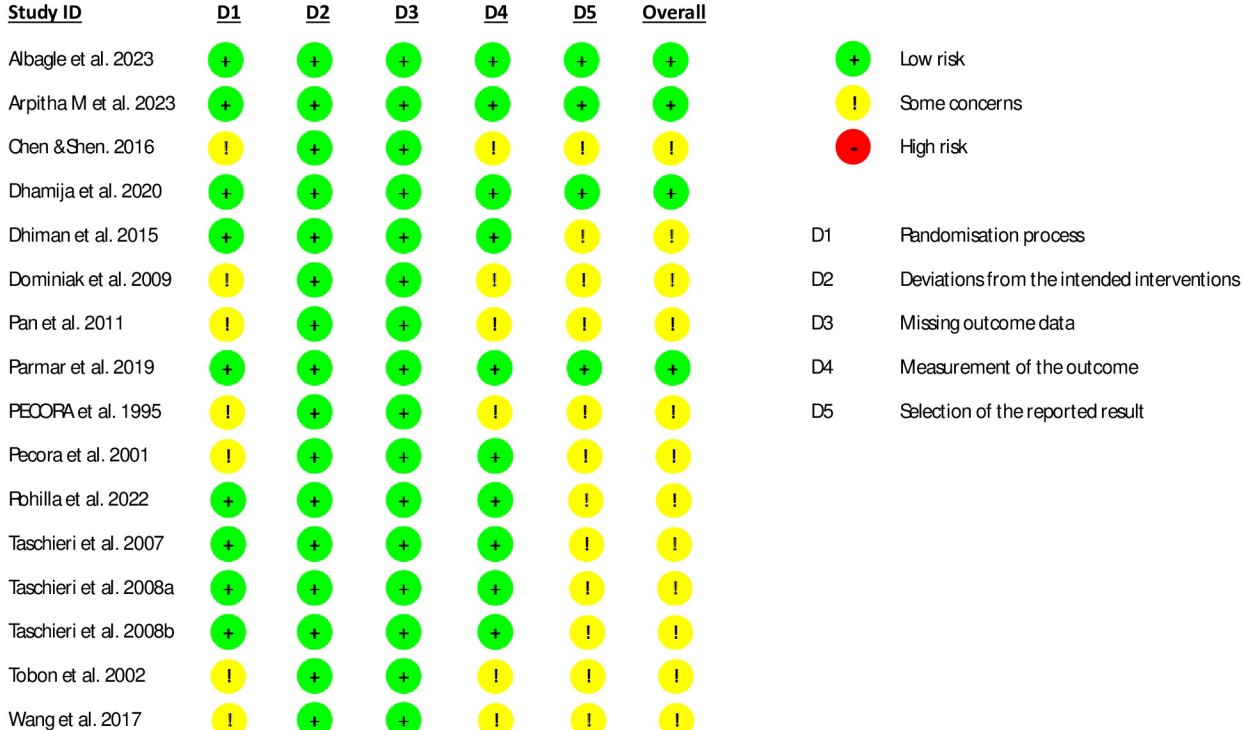

**Fig 2. Risk of bias assessment.**

0.34–0.73; P < 0.001) (Fig 3). Four studies [50, 51, 53, 57] reported the failure rate according to 3D evaluation. The results showed that using GTR following SRCT significantly improved the healing process compared to the conventional SRCT (RR: 0.36; 95% CI, 0.15–0.90; P < 0.001) (Fig 4).

## Subgroup analysis based on the techniques and materials

When e-PTFE membranes were used alone, no significant effects were seen in the healing process following SRCT (RR: 2.00; 95% CI, 0.22–18.33; P = 0.54). When resorbable collagen membranes were used alone, the results showed better outcomes but were statistically not significantly different from the control group (RR: 0.66, 95% CI: 0.29–1.52, p = 0.33). In groups that received only bone grafts or APCs, outcomes were also slightly better but not significantly different (bone grafts: RR: 0.59; 95% CI, 0.27–1.27; P = 0.17; APCs: RR: 0.75; 95%, 0.19–3.02; P = 0.69). When bovine bone-derived hydroxyapatite with collagen membrane was used in combination, the success rate was significantly increased compared to the control group (RR: 0.43, 95% Cl: 0.25–0.74, p < 0.001) (Fig 5).

## Subgroup analysis based on the lesion type

Six studies utilized GTR procedures on confined periapical lesions; there was an advantage towards improved outcomes in GTR groups, without statistically significant difference (RR: 0.59; 95% CI, 0.34–1.02; P = 0.06). When GTR is used on the apico-marginal defect with complete root exposure on the buccal side, there is no significant impact on the healing process after SRCT (RR: 2.00; 95% CI, 0.22–18.33; P = 0.54). When the GTR procedures were used on patients with through-and-through lesions, the results indicated a significant increase in the

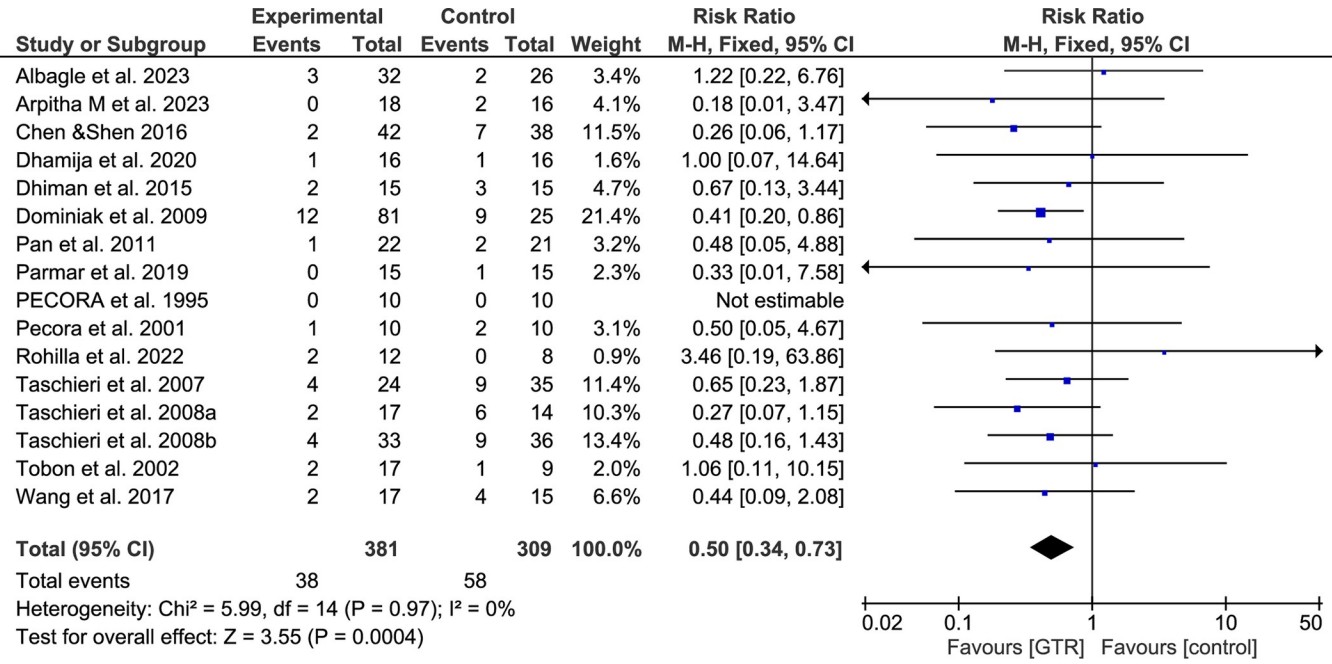

**Fig 3. Forest plot and meta-analysis results for 2D evaluation.** "Events" indicate failure cases.

success rate of the GTR group compared to the control group (RR: 0.36, 95% Cl: 0.19–0.68, p < 0.001) (Fig 6).

## Publication bias

A funnel plot was created to evaluate the presence of publication bias. The findings indicated that the funnel plot showed bilateral symmetry, indicating the absence of notable publication bias (Fig 7).

## Certainty of evidence

We applied the GRADE process to rank the confidence level of the evidence obtained through our meta-analysis that evaluated the effect of GTR procedures on SRCT (Table 2). After considering the serious risk of bias domain and the non-serious indirectness, imprecision, and inconsistency domains, the success rate of GTR procedures following SRCT is considered to have a moderate grade of evidence.

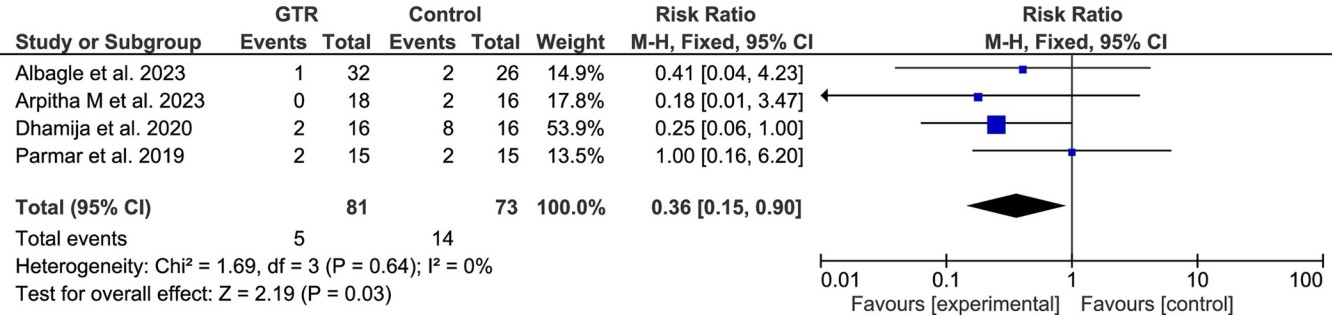

**Fig 4. Forest plot and meta-analysis results for 3D evaluation.** "Events" indicate failure cases.

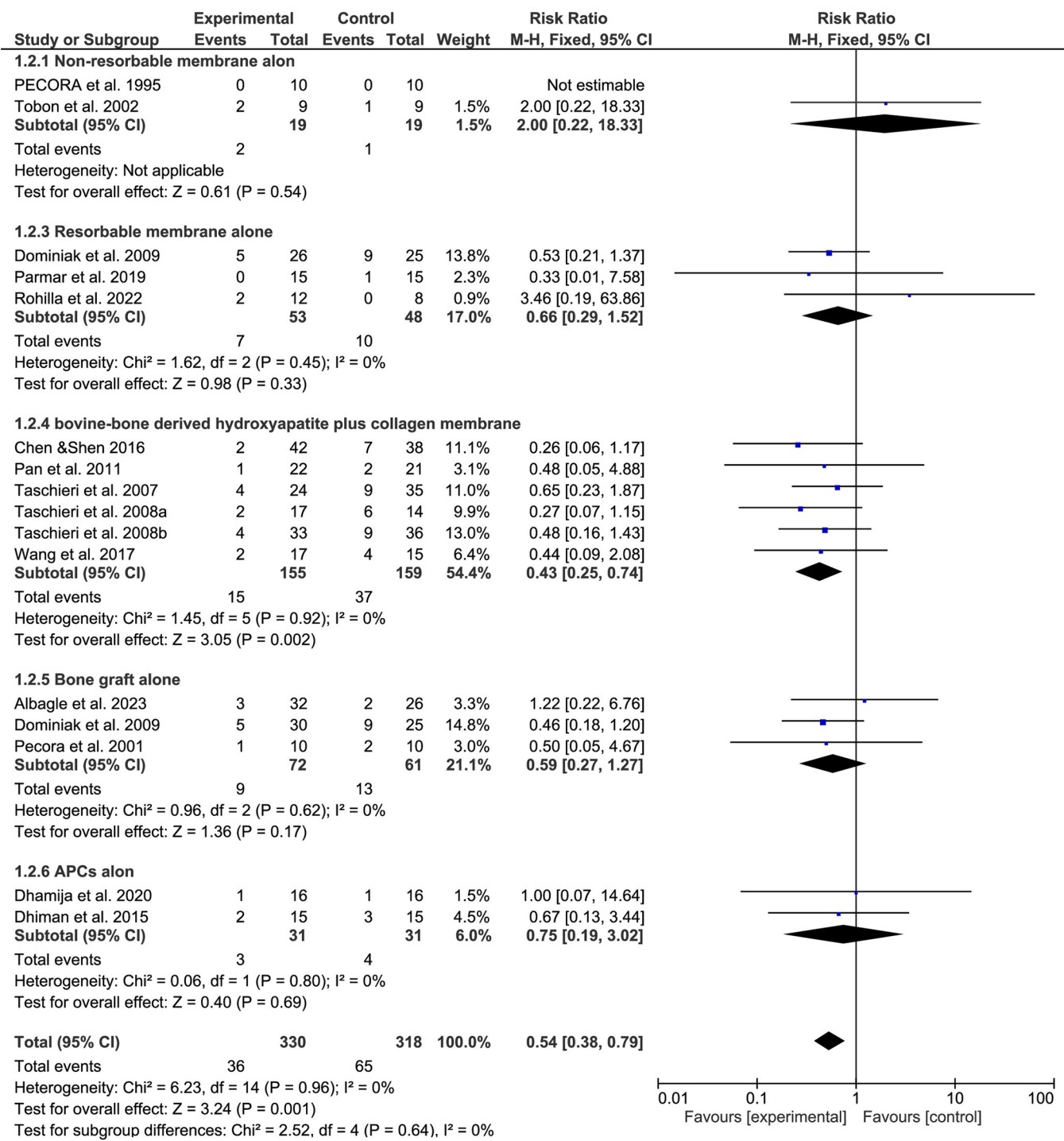

**Fig 5. Forest plot and Subgroup analysis based on the GTR technique and materials.** "Events" indicate failure cases.

## Discussion

GTR procedures have been used to enhance bone regeneration in SRCT [66]. However, different techniques and biomaterials lead to conflicting outcomes [67], which are still controversial. The current systematic review and meta-analysis include RCTs, which aim to investigate the

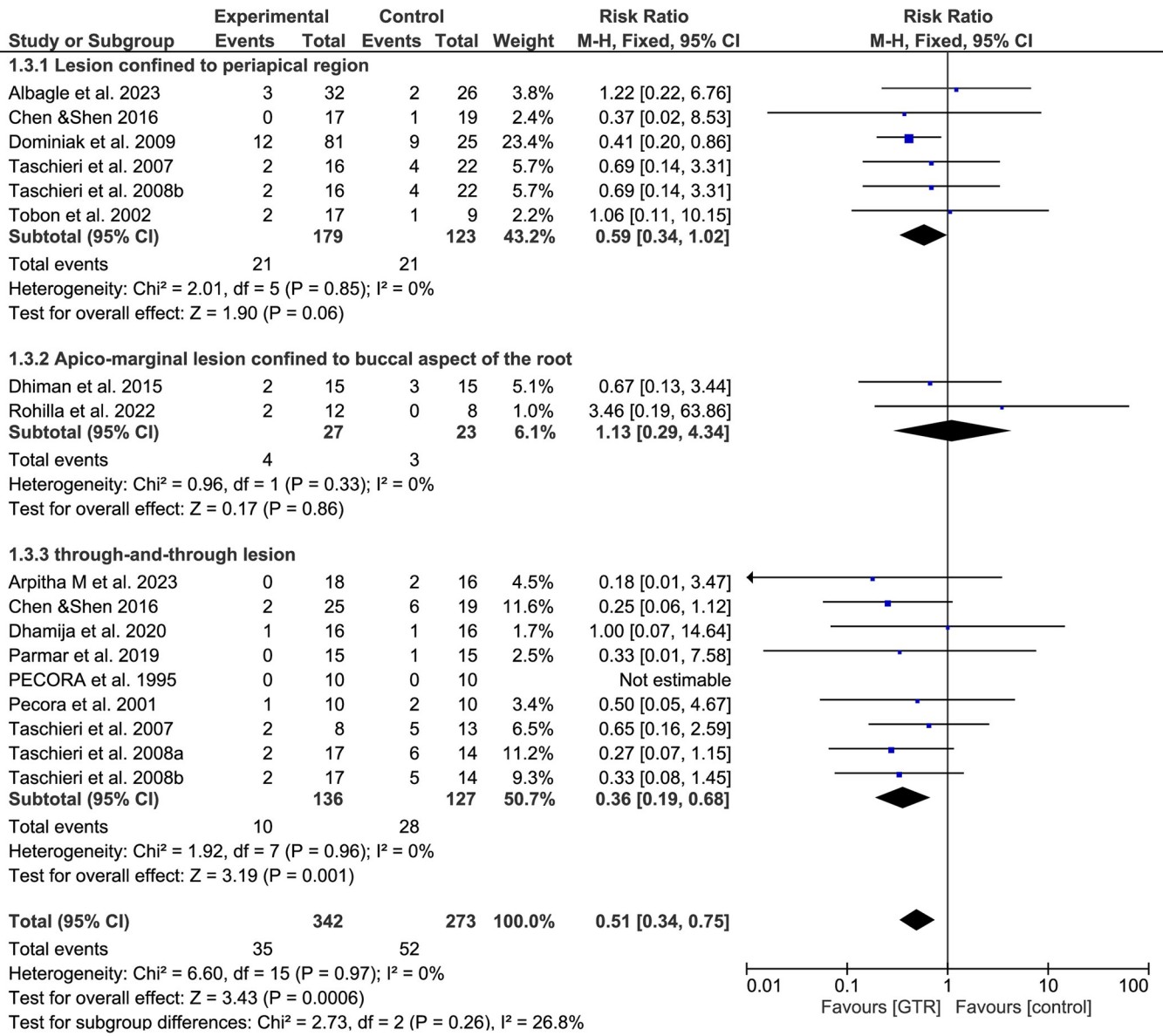

**Fig 6. Forest plot and subgroup analysis based on lesion type.** "Events" indicate failure cases.

effects of GTR procedures on the outcome of SRCT. Because there is no outcome difference between the one-year and four-year follow-ups, the studies with more than a one-year follow-up were reported, and the data was explicitly extracted based on the one-year follow-up [68]. Still, it is essential to be careful when generalizing the study's findings regarding the long-term surgical outcomes of SRCT. The current meta-analysis shows that GTR procedures significantly improve the healing process one year after SRCT, regardless of the evaluation method used. However, results varied in subgroup analysis. Using e-PTFE membranes alone did not improve outcomes, but using collagen membranes, bone grafts, or APCs alone may accelerate healing. Using collagen membranes and bovine bone-derived hydroxyapatite together showed significant improvements. Furthermore, results varied according to the lesion type. GTR treatment for apico-marginal lesions did not improve outcomes, but may accelerate the healing in

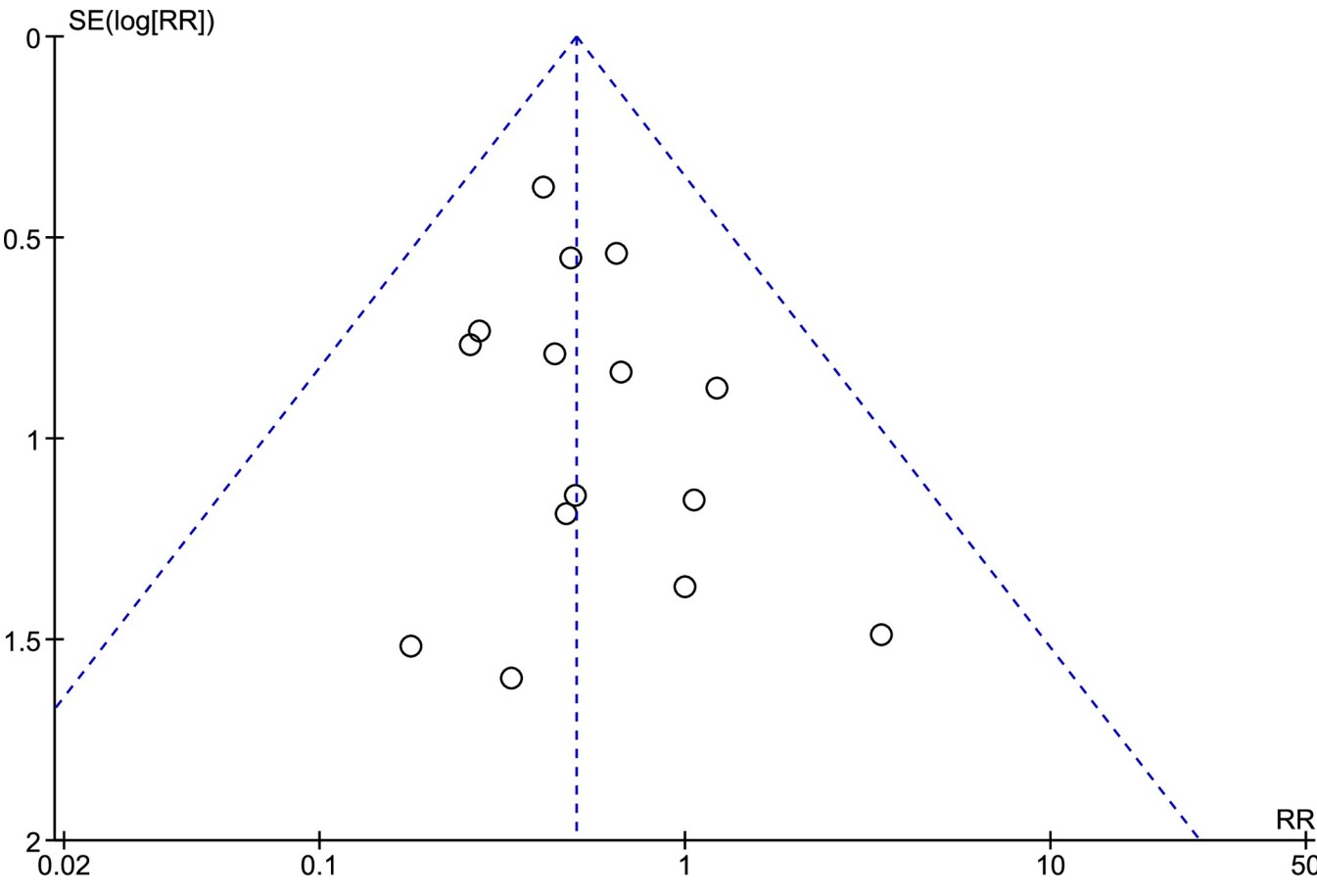

**Fig 7. The funnel plot of publication bias.**

confined lesions and significantly improve the healing in through-and-through lesions. This meta-analysis was conducted on various types of permanent teeth with different lesion sizes, focusing on lesions caused by endodontic problems. Therefore, the results may apply to patients of different ages with permanent teeth. However, they may not apply to patients with combined Endo-Perio lesions.

The first barrier membrane employed in SRCT as a GTR technique was e-PTFE [69]. This membrane prevents soft tissue growth inside defective areas following SRCT and promotes osteoblast development [70]. However, only two studies [59, 64] in our analysis used this approach, and no evidence justifies its use in SRCT. In contrast, Yoshikawa et al. [71] conducted a study on beagle dogs. They found that using the e-PTFE membrane significantly increased the formation of new cortical bone compared to the control group. Furthermore, using e-PTFE may lead to complications such as membrane exposure and bacterial infection [72] because it is non-absorbable and requires surgical extraction, which increases patient suffering, treatment expenses, and possible issues [73]. The e-PTFE membrane was recently withdrawn from use in SRCT. However, there are alternate materials available [74].

Collagen membranes are the later generation of absorbable membranes and offer several advantages over non-absorbable membranes, including cost-effectiveness and reduced risk of complications [69, 73]. In the present study, absorbable collagen membranes were included in 3 RCTs [55, 57, 60], and no significant differences were observed compared to the control group. Our findings support using collagen membranes, with no statistically significant

**Table 2. Certainty of evidence according to GRADE.**

**[GTR] compared to [non-GTR] for [SRCT]**

**Patient or population:** [patients having SRCT]

**Intervention:** [GTR]

**Comparison:** [non-GTR]

| Outcomes | Anticipated absolute effects* (95% CI) | | Relative effect (95% CI) | of participants (studies) | Certainty of the evidence (GRADE) | Comments |
|---|---|---|---|---|---|---|
| | Risk with [non-GTR] | Risk with [GTR] | | | | |
| Healing process after SRCT | 19 per 100 | **9 per 100** | **RR 0.50** | 690 | | The GTR procedure likely reduces the failure rate following SRCT. |
| | | (6 to 7) | (0.34 to 0.73) | (16 RCTs) | Moderate[a,b,c,d] | |

*__The risk in the intervention group__ (and its 95% confidence interval) is based on the assumed risk in the comparison group and the __relative effect__ of the intervention (and its 95% CI).

**CI:** Confidence Interval; **RR:** Risk Ratio

**GRADE Working Group grades of evidence**

**High certainty:** We are very confident that the true effect lies close to that of the estimate of the effect.

**Moderate certainty:** we are moderately confident in the effect estimate; the true effect is likely to be close to the estimate of the effect, but there is a possibility that it is substantially different.

**Low certainty:** our confidence in the effect estimate is limited: the true effect may be substantially different from the estimate of the effect.

**Very low certainty:** we have very little confidence in the effect estimate: the true effect is likely to be substantially different from the estimate of effect.

a, 'Som concern" risk of bias with allocation concealment, measurement of outcome, and selective reporting; B, Visual inconsistency and statistical analysis show no heterogeneity; c, Direct comparison; d, Narrow confidence interval.

differences observed compared to the control group. Our finding is consistent with Dominika et al. [55], who also observed a higher success rate after 6 months of follow-up when using collagen membranes. However, after 12 months, there was no significant difference in success rates. Their study showed that using collagen membranes can accelerate the healing process after SRCT. The collagen membrane has varying resorption times, and it is essential for the optimal resorption time to match the time for bone regeneration. Cross-linked technology has been utilized to extend the resorption time and facilitate successful healing [75]. Hence, the collagen membrane should remain in place for a long time.

Using Collagen membranes alone can be unstable, which causes collapse under loads and delayed bone tissue regeneration [76]. To address this issue, some authors have used bone grafts [77]. Bovine bone-derived hydroxyapatite is commonly used because it is a biocompatible graft material with osteoconductive characteristics [78]. The slow resorption rate of bovine bone-derived hydroxyapatite is a critical advantage, as it enables better integration and acts as an effective osteoconductive grafting material during the natural healing process. This can ultimately lead to successful bone healing outcomes [79]. In this study, 6 RCTs [52, 55, 61–63, 65] utilized collagen membranes and bovine bone-derived hydroxyapatite grafts, and we found that this combination significantly improved healing following SRCT. Similar results were reported by Wang et al.[65], who found a significantly better success rate after 12 months of follow-up using collagen membranes combined with bovine bone-derived hydroxyapatite graft.

The bone graft is the most common GTR procedure utilized [12]. In this study, three studies [50, 55, 58] used three different types of bone grafts without additional materials, and we found that this may result in improved healing following SRCT, but without a statistically significant difference. However, Sreedevi et al. [80] found that using freeze-dried hydroxyapatite bone graft material resulted in successful bone healing compared to the control group. The

different types of bone grafts have varying success levels and potential advantages and disadvantages [12]. Autogenous bone grafts are the only type that has osteogenesis, osteoinduction, and osteoconduction properties, with a success rate of 95% [15]; however, due to their drawbacks, such as postoperative pain, longer surgery time, and increased morbidity, other graft types are often preferred, such as allograft, xenograft, and alloplastic grafts [81]. Nonetheless, some studies have shown that combining bone grafts with barrier membranes can improve clinical outcomes [82].

Some studies have investigated using APCs to improve healing and regeneration after SRCT [12]. The APCs provide platelets, leukocytes, and growth factors that promote tissue growth and blood flow [83]. Our current meta-analysis found that using APCs resulted in more favorable healing outcomes without a statistically significant difference. Dhamija et al. [53] found a significantly better success rate with PRP in a 3D assessment but no significant difference in a 2D evaluation. The i-PRF is the third generation of APCs [25]. In this analysis, only one study [51] used i-PRF with a collagen-based bone graft. Further research is needed to establish the efficacy of this treatment protocol.

SRCT may involve dealing with many compromised conditions. Based on several clinical and experimental studies, periapical lesions have been categorized into three main types: (1) confined to periapical areas without erosion of the lingual cortex, (2) through-and-through lesions (tunnel), and (3) apico-marginal lesions [4].

In this study, 6 clinical trials [50, 52, 55, 61, 63, 64] assessed several GTR procedures to promote the repair of a confined lesion in the periapical area. Our findings indicate that utilizing GTR procedures can accelerate wound healing, although there is no significant difference in situations of confined periapical lesions. Our finding is consistent with Chen & Shen [52]. They found that the GTR group demonstrated a better success rate than the control group after 6 months, but the difference was not statistically significant after a year.

Apico-marginal lesions can cause epithelial down growth across the denuded root surface after SRCT, which increases the possibility of a recurrent connection between the apical and marginal tissues. Only two studies [54, 60] in our analysis used GTR procedures for apico-marginal lesions with complete exposure on the buccal surface of the root. Our findings indicate that regenerative techniques may not significantly impact outcomes for these lesions. However, a 2023 case series by Baruwa et al. [84] found that endodontic microsurgery combined with GTR can be a highly effective treatment approach for treating apico-marginal lesions. It is important to note that proper diagnosis and procedures are crucial for achieving successful results. Furthermore, as the interest in using APCs for apico-marginal defects grows, better evidence is needed to support their effectiveness. Further well-conducted trials are necessary to fully understand the potential impact of APCs in these cases.

Fast soft tissue proliferation from the facial and lingual sides can hinder bone growth and lead to incomplete healing or scar tissue formation in through-and-through lesions [4]. Furthermore, these through-and-through lesions may offer a pathway for bacterial infection. Therefore, the inclusion of GTR procedures not only have a regenerative function but also play a crucial role in blocking this pathway [51]. This meta-analysis included 9 studies [51–53, 57–59, 61–63] that evaluated the effects of GTR procedures for these lesions and found that they significantly improve wound healing following SRCT. Similar results were reported by Taschieri et al. [63] when using collagen membrane and hydroxyapatite from bovine bone.

## Strength and limitation

This systematic review and meta-analysis only contained RCTs, which provide robust and reliable evidence. Comprehensive inclusion and exclusion criteria were applied to ensure a

focused research question and avoid bias in article selection [85]. The search included multiple databases without language or geographical restrictions, increasing the potential for generalization [86]. The study also considered the impact of lesion type on the effectiveness of different GTR procedures. Subgroup analysis can provide clinicians with further insights into choosing the suitable GTR procedure for different lesion types.

There are various limitations to the current study. Most studies raised "some concerns" based on the Cochrane Collaboration tool. In addition, only a single research study incorporated the third generation of APCs. Therefore, meticulously planned clinical trials of superior quality are necessary. Also, only four studies were included that used modified PENN 3D criteria to evaluate the healing process after SRCT. In recent years, CBCT has been used in more clinical studies by using different criteria. More research needs to be carried out to develop straightforward, standardized ways to evaluate the results of 3D radiographs.

## Conclusion

This meta-analysis showed that GTR processes significantly improve healing after SRCT, regardless of evaluation methods, especially when collagen membranes and bovine bone-derived hydroxyapatite are used together. Furthermore, for through-and-through lesions, the GTR procedures significantly improved the healing after SRCT.

## Supporting information

**S1 Checklist. PRISMA 2020 checklist.**
(DOCX)

**S1 Table. Search strategy.**
(DOCX)

**S2 Table. Studies identified after excluding duplications.**
(XLSX)

**S3 Table. The excluded studies and reasons for exclusion.**
(DOCX)

**S4 Table. All data extracted in primary studies.**
(XLSX)

**S5 Table. Completed risk of bias assessments.**
(XLSX)

## Author Contributions

**Conceptualization:** Nader Muthanna, Ang Li.

**Data curation:** Nader Muthanna, Xiaoyue Guan, Fouad Alzahrani, Badr Sultan Saif, Abdelrahman Seyam, Ahmed Alsalman, Ahmed Es Alajami, Ang Li.

**Formal analysis:** Nader Muthanna, Xiaoyue Guan, Badr Sultan Saif.

**Funding acquisition:** Ang Li.

**Investigation:** Nader Muthanna, Xiaoyue Guan, Fouad Alzahrani, Badr Sultan Saif, Abdelrahman Seyam, Ahmed Alsalman, Ahmed Es Alajami, Ang Li.

**Methodology:** Nader Muthanna, Xiaoyue Guan, Fouad Alzahrani, Badr Sultan Saif, Abdelrahman Seyam, Ahmed Alsalman, Ahmed Es Alajami, Ang Li.

**Project administration:** Ang Li.

**Resources:** Ang Li.

**Software:** Nader Muthanna, Xiaoyue Guan, Fouad Alzahrani, Badr Sultan Saif.

**Supervision:** Ang Li.

**Validation:** Nader Muthanna, Xiaoyue Guan, Ang Li.

**Visualization:** Ang Li.

**Writing – original draft:** Nader Muthanna, Xiaoyue Guan, Ang Li.

**Writing – review & editing:** Nader Muthanna, Xiaoyue Guan, Ang Li.

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
