## [Decision Letter · Decision Letter 0]

21 Aug 2024

PONE-D-24-25494Impact of Regenerative Procedure on The Healing Process Following Surgical Root Canal Treatment: A systematic review and meta-analysisPLOS ONE

Dear Dr. Hussain,

Thank you for submitting your manuscript to PLOS ONE. After careful consideration, we feel that it has merit but does not fully meet PLOS ONE’s publication criteria as it currently stands. Therefore, we invite you to submit a revised version of the manuscript that addresses the points raised during the review process.

I have identified some significant issues that need to be addressed before the paper can be considered for publication.

The main concerns are as follows:

Numerous grammatical errors throughout the manuscriptMultiple spelling mistakesInconsistent formatting

Given these issues, I am recommending a Major Revision for your paper. In addition to addressing the reviewers' comments, please pay careful attention to the following:

Thoroughly proofread the entire manuscript for grammatical and spelling errors.Ensure consistent formatting throughout the paper.Review all tables carefully for spelling, formatting, and accuracy.

I encourage you to consider using a professional proofreading service if needed. Addressing these issues will significantly improve the quality and readability of your work.

Please submit your revised manuscript along with a point-by-point response to the reviewers' comments and a detailed list of the changes you've made to address the language and formatting issues

We look forward to receiving your revised manuscript.

Kind regards,

Fahad Umer

Academic Editor

PLOS ONE

Journal Requirements:

2. We note that your Data Availability Statement is currently as follows: "All relevant data are within the manuscript and its Supporting Information files."

Additional Editor Comments:

Dear Authors,

I hope this email finds you well. I have received two reviews for your manuscript submitted to PLOS ONE. While both reviewers have suggested only minor changes to the content, I have identified some significant issues that need to be addressed before the paper can be considered for publication.

The main concerns are as follows:

Numerous grammatical errors throughout the manuscript

Multiple spelling mistakes

Inconsistent formatting

Given these issues, I am recommending a Major Revision for your paper. In addition to addressing the reviewers' comments, please pay careful attention to the following:

Thoroughly proofread the entire manuscript for grammatical and spelling errors.

Ensure consistent formatting throughout the paper.

Review all tables carefully for spelling, formatting, and accuracy.

I encourage you to consider using a professional proofreading service if needed. Addressing these issues will significantly improve the quality and readability of your work.

Please submit your revised manuscript along with a point-by-point response to the reviewers' comments and a detailed list of the changes you've made to address the language and formatting issues.

If you have any questions or need clarification on any point, please don't hesitate to contact me.

Best regards,

Fahad

Reviewers' comments:

Reviewer's Responses to Questions

**Comments to the Author**

1. Is the manuscript technically sound, and do the data support the conclusions?

Reviewer #1: Yes

Reviewer #2: Yes

2. Has the statistical analysis been performed appropriately and rigorously? 

Reviewer #1: Yes

Reviewer #2: Yes

3. Have the authors made all data underlying the findings in their manuscript fully available?

Reviewer #1: Yes

Reviewer #2: Yes

4. Is the manuscript presented in an intelligible fashion and written in standard English?

Reviewer #1: Yes

Reviewer #2: Yes

5. Review Comments to the Author

Reviewer #1: The authors are commended for the thought process and a well written scientific manuscript. The current systematic review will be of good interest to the scientific community.

There are some minor details that need modification to improve the overall quality of the manuscript.

1. There are several punctuation errors and some typographical errors all through the manuscript. The use of full stop, comma, semi colon and numbers are used in sentences which changed the context entirely. Advise is to seek a professional English language reviewer.

2. The formatting of the reference should be checked to ensure it matches the guidelines of the journal.

3. The last sentence of the paragraph states "However, The GTR procedure has no advantage in treating apical marginal lesions." - This statement may not be entirely true as the application of GTR in cases of marginal defects is dependent on the diagnosis and appropriate use of GTR technique. See the case series published in JOE, although a low level of evidence, it provides insight on the use of GTR in apical marginal lesions https://doi.org/10.1016/j.joen.2023.07.009

Reviewer #2: Introduction:

GTR is comprehensively covered. Add more information about nonsurgical root canals and surgical root canal assessment criteria. 

Methods:

Data Extraction: Give evidence for the criteria that you used to assess success and failure. 

Discussion: 

2nd paragraph: What you met by possible issues?

You did not discuss the slow resorption rate of bovine bone-derived hydroxyapatite, or whether it could affect the radiographic findings.

6. PLOS authors have the option to publish the peer review history of their article (what does this mean?). If published, this will include your full peer review and any attached files.

Reviewer #1: No

Reviewer #2: No

---

## [Author Response · Author response to Decision Letter 0]

20 Sep 2024

Response to Reviewers

Prof. Fahad Umer

Academic Editor 

PLOS ONE

20 September 2024

Dear Academic Editor [Fahad Umer]

Subject: Submission of revised manuscript No.: PONE-D-24-25494

Impact of Regenerative Procedure on The Healing Process Following Surgical Root Canal Treatment: A systematic review and meta-analysis

We appreciate the opportunity to submit a revised revision of our manuscript. It is greatly appreciated that you and the other reviewers have devoted the time and effort necessary to offer valuable feedback on our manuscript. Also, we thank the reviewers for their insightful feedback regarding our paper. We have successfully implemented modifications to incorporate most of the reviewers' recommendations. We highlighted the modifications in the Revised Manuscript by the red color. 

We hope the revised version is appropriate for publishing and look forward to your response.

Additional Editor Comments:

The main concerns are as follows:

Numerous grammatical errors throughout the manuscript

Multiple spelling mistakes

Inconsistent formatting

Response:

Thank you for raising these points. We have already hired a professional proofreader to thoroughly review all sections of the paper as recommended. 

Response to Reviewer #1:

Thank you for taking the time to review our paper. Your feedback and suggestions have been valuable in improving the quality of our work. 

We have carefully addressed your point and made the necessary modifications as suggested.

1. There are several punctuation errors and some typographical errors all through the manuscript. The use of full stop, comma, semi colon and numbers are used in sentences which changed the context entirely. Advice is to seek a professional English language reviewer.

Response:

Thank you for raising this point. We have already hired a professional proofreader to thoroughly review all sections of the paper as recommended.

2. The formatting of the reference should be checked to ensure it matches the guidelines of the journal.

Response:

Thank you for raising this point. We have re-checked and downloaded the referencing style from the PLoS ONE website to ensure it matches the journal's guidelines.

3. The last sentence of the paragraph states "However, The GTR procedure has no advantage in treating apical marginal lesions." - This statement may not be entirely true as the application of GTR in cases of marginal defects is dependent on the diagnosis and appropriate use of GTR technique. See the case series published in JOE, although a low level of evidence, it provides insight on the use of GTR in apical marginal lesions https://doi.org/10.1016/j.joen.2023.07.009

Response:

Thank you for your valuable suggestions. This sentence was written in the conclusion, so it is crucial to ensure that the conclusion accurately reflects the study's key findings. We have included this point in the discussion part as recommended. (P.15, Line 250).

[However, a 2023 case series by Baruwa et al. found that endodontic microsurgery combined with GTR can be a highly effective treatment approach for treating apico-marginal lesions. It is important to note that proper diagnosis and procedures are crucial for achieving successful results.]

Response to Reviewer #2:

Thank you for taking the time to review our paper. Your feedback and suggestions have been valuable in improving the quality of our work. 

We have carefully addressed your points and made the necessary modifications as suggested.

1. Introduction: GTR is comprehensively covered. Add more information about nonsurgical root canals and surgical root canal assessment criteria.

Response:

Thank you for your valuable suggestions. We have included this point in the introduction part as recommended. (p.3, line 27).

[Success in both RCT and SRCT relies on the absence of signs of infection and inflammation, along with radiography showing reduced periapical lesion size and normal growth of the periodontal ligament gap. The evaluation of healing after SRCT is commonly conducted using the criteria established by Rud et al. and Molven et al. on 2D imaging, which categorizes healing as complete, incomplete, uncertain, or unsatisfactory. On the other hand, the Modified PENN criteria have been used to evaluate healing on 3D imaging.]

2. Methods: Data Extraction: Give evidence for the criteria that you used to assess success and failure.

Response:

Thank you for raising this point. The success and failure were established based on the criteria described by Rud et al. and Molven et al. for 2D imaging and modified PENN's criteria described by Schloss et al. for 3D imaging. For statistical purposes, the outcomes were dichotomized into success and failure; this method of dichotomization has been described and used in many Randomized control trials. We clarified that in the Data Extraction section as recommended. (p.6, line 95).

Here are some references that use the same dichotomization method: Schloss et al.[1], Parmer et al.[2], Dhamija et al.[3]

[The clinical outcomes are evaluated by the presence or absence of signs of infection and inflammation. Radiographically, the healing assessment was determined by using the criteria established by Rud et al. or Molven et al. (complete, incomplete, uncertain, or unsatisfactory healing) for 2D imaging evaluation, whereas the modified PENN criteria established by Schloss et al. (complete, limited, uncertain, or unsatisfactory healing) were used for 3D imaging evaluation.

The assessment of success and failure was determined based on a comprehensive evaluation of both clinical and radiological outcomes. For statistical purposes, the outcomes were also dichotomized into success and failure. The Success was assessed by the loss of clinical symptoms and the signs of (complete or incomplete healing) for 2D imaging and (Complete or Limited healing) for 3D imaging. Failure was assessed by the presence of clinical symptoms and/or the occurrence of (uncertain or unsatisfactory healing) for 2D and 3D imaging.]

3. Discussion: 2nd paragraph: What you met by possible issues?

You did not discuss the slow resorption rate of bovine bone-derived hydroxyapatite, or whether it could affect the radiographic findings.

Response:

Thank you for raising this point. This point has been added to the discussion part as recommended. (p.14, line 217).

[The slow resorption rate of bovine bone-derived hydroxyapatite is a critical advantage, as it enables better integration and acts as an effective osteoconductive grafting material during the natural healing process. This can ultimately lead to successful bone healing outcomes.]

References

1. Schloss T, Sonntag D, Kohli MR, Setzer FC. A Comparison of 2- and 3-dimensional Healing Assessment after Endodontic Surgery Using Cone-beam Computed Tomographic Volumes or Periapical Radiographs. J Endod. 2017;43(7):1072-9. Epub 20170517. doi: 10.1016/j.joen.2017.02.007. PubMed PMID: 28527841.

2. Parmar PD, Dhamija R, Tewari S, Sangwan P, Gupta A, Duhan J, Mittal S. 2D and 3D radiographic outcome assessment of the effect of guided tissue regeneration using resorbable collagen membrane in the healing of through-and-through periapical lesions - a randomized controlled trial. Int Endod J. 2019;52(7):935-48. Epub 20190306. doi: 10.1111/iej.13098. PubMed PMID: 30758848.

3. Dhamija R, Tewari S, Sangwan P, Duhan J, Mittal S. Impact of Platelet-rich Plasma in the Healing of Through-and- through Periapical Lesions Using 2-dimensional and 3-dimensional Evaluation: A Randomized Controlled Trial. Journal of Endodontics. 2020;46(9):1167-84. doi: 10.1016/j.joen.2020.06.004. PubMed PMID: WOS:000572345400002.

---

## [Editor Report · Decision Letter 1]

4 Oct 2024

PONE-D-24-25494R1Impact of Regenerative Procedure on The Healing Process Following Surgical Root Canal Treatment: A systematic review and meta-analysisPLOS ONE

Dear Dr. Hussain

Thank you for submitting your manuscript to PLOS ONE. After careful consideration, we feel that it has merit but does not fully meet PLOS ONE’s publication criteria as it currently stands. Therefore, we invite you to submit a revised version of the manuscript that addresses the points raised during the review process.

Please follow my comments  

Editors comments

**Introduction:**

It’s a bit prolix and can benefit from making it concise

**Statistical analysis**

Random effect model was not used so dis-include this.

**Results :**

Fig 1 : Prisma 2020

Use this flow chart RISMA 2020 flow diagram for new systematic reviews which included searches of databases, registers and other sources (https://www.prisma-statement.org/prisma-2020-flow-diagram) be mindful of formatting and font sizes.

Table 1 is not required since it will be in Prisma flow diagram and will be redundant information.

Discussion

PENN is misspelled as PEEN

Please pay close attention to these details and resubmit

Conclusion

Modify the conclusion to focus on the statistically significant GTR techniques  

We look forward to receiving your revised manuscript.

Kind regards,

Fahad Umer

Academic Editor

PLOS ONE

---

## [Author Response · Author response to Decision Letter 1]

10 Oct 2024

Response to Reviewers

Prof. Fahad Umer

Academic Editor 

PLOS ONE

10 October 2024

Dear Academic Editor [Fahad Umer]

Subject: Submission of revised manuscript No.: PONE-D-24-25494R1

Impact of Regenerative Procedure on The Healing Process Following Surgical Root Canal Treatment: A systematic review and meta-analysis

We appreciate the opportunity to submit a revised revision of our manuscript. It is greatly appreciated that you and the other reviewers have devoted the time and effort necessary to offer valuable feedback on our manuscript. We have successfully implemented modifications to incorporate all of the Editor's recommendations. We utilized the track changes feature in the Word file of the revised manuscript. 

We hope the revised version is appropriate for publishing and look forward to your response.

Editor Comments:

Introduction:

1. It’s a bit prolix and can benefit from making it concise

Response:

Thank you for your suggestion regarding the text's verbosity. We have addressed this issue and made the necessary revisions to ensure the introduction is as concise as recommended. (P.3,4).

Statistical analysis

2. Random effect model was not used, so dis-include this.

Response:

Thank you for raising this point. we have noted your suggestion and have removed any references to random effect as recommended. (P.7, Line 223).

Results:

Fig 1: Prisma 2020

3. Use this flow chart RISMA 2020 flow diagram for new systematic reviews which included searches of databases, registers and other sources (https://www.prisma-statement.org/prisma-2020-flow-diagram) be mindful of formatting and font sizes.

Response:

 Thank you for your valuable suggestions. We used the RISMA 2020 flow diagram for new systematic reviews, which included searches of databases, registers, and other sources as recommended.

4. Table 1 is not required since it will be in the Prisma flow diagram and will be redundant information.

Response: 

Thank you for your feedback regarding Table 1. We have removed it from the main manuscript as recommended. Instead, we have included it as supporting information. (P.8).

Discussion

5. PENN is misspelled as PEEN

Response: Thank you for pointing out the spelling error. We have corrected "PEEN" to "PENN" throughout the manuscript. We appreciate your attention to detail. (P.16, Line 285).

Conclusion

6. Modify the conclusion to focus on the statistically significant GTR techniques 

 Response:

Thank you for your suggestion. We revised and rewrote the conclusion to emphasize the key statistically significant impacts of GTR procedures. (P.16, Line 289).

---

## [Editor Report · Decision Letter 2]

14 Oct 2024

Impact of Regenerative Procedure on The Healing Process Following Surgical Root Canal Treatment: A systematic review and meta-analysis

PONE-D-24-25494R2

Dear Dr. Hussain,

We’re pleased to inform you that your manuscript has been judged scientifically suitable for publication and will be formally accepted for publication once it meets all outstanding technical requirements.

Kind regards,

Fahad Umer

Academic Editor

PLOS ONE
---

## [Editor Report · Acceptance letter]

18 Dec 2024

PONE-D-24-25494R2 

PLOS ONE

Dear Dr. Li, 

I'm pleased to inform you that your manuscript has been deemed suitable for publication in PLOS ONE. Congratulations! Your manuscript is now being handed over to our production team.

Kind regards, 

on behalf of

Dr. Fahad Umer 

Academic Editor

PLOS ONE